# Resonant Inner-Shell Photofragmentation of Adamantane (C10H16)

**DOI:** 10.3390/molecules28145510

**Published:** 2023-07-19

**Authors:** Smita Ganguly, Mathieu Gisselbrecht, Per Eng-Johnsson, Raimund Feifel, Sergio Díaz-Tendero, Eva Muchová, Aleksandar R. Milosavljević, Patrick Rousseau, Sylvain Maclot

**Affiliations:** 1Department of Physics, Lund University, 22100 Lund, Sweden; smita_omkarnath.ganguly@sljus.lu.se (S.G.); mathieu.gisselbrecht@sljus.lu.se (M.G.); per.eng-johnsson@fysik.lth.se (P.E.-J.); 2Department of Physics, University of Gothenburg, Origovagen 6 B, 41296 Gothenburg, Sweden; raimund.feifel@physics.gu.se (R.F.); sylvain.maclot@univ-lyon1.fr (S.M.); 3Department of Chemistry, Universidad Autonoma de Madrid, 28049 Madrid, Spain; sergio.diaztendero@uam.es; 4Institute for Advanced Research in Chemistry (IAdChem), Universidad Autónoma de Madrid, 28049 Madrid, Spain; 5Condensed Matter Physics Center (IFIMAC), Universidad Autónoma de Madrid, 28049 Madrid, Spain; 6Department of Physical Chemistry, University of Chemistry and Technology, Technická 5, 166 28 Prague, Czech Republic; eva.muchova@vscht.cz; 7Synchrotron SOLEIL, L’Orme de Merisiers, Saint Aubin, BP48, 91192 Gif-sur-Yvette CEDEX, France; 8Normandie University, ENSICAEN, UNICAEN, CEA, CNRS, CIMAP, 14000 Caen, France; prousseau@ganil.fr; 9Institut Lumiere Matiere UMR 5306, Universite Claude Bernard Lyon 1, CNRS, Univ. Lyon, 69100 Villeurbanne, France

**Keywords:** adamantane, inner-shell fragmentation, site-selectivity, AE–PICO/PIPICO coincidence

## Abstract

Adamantane, the smallest diamondoid molecule with a symmetrical cage, contains two distinct carbon sites, CH and CH${}_{2}$. The ionization/excitation of the molecule leads to the cage opening and strong structural reorganization. While theoretical predictions suggest that the carbon site CH primarily causes the cage opening, the role of the other CH${}_{2}$ site remains unclear. In this study, we used advanced experimental Auger electron–ion coincidence techniques and theoretical calculations to investigate the fragmentation dynamics of adamantane after resonant inner-shell photoexcitation. Our results demonstrate that some fragmentation channels exhibit site-sensitivity of the initial core–hole location, indicating that different carbon site excitations could lead to unique cage opening mechanisms.

## 1. Introduction

Synchrotron radiation allows resonant excitation of a specific atomic site in molecules. Numerous studies of fragmentation of core-excited molecules have found site-selective fragmentation [1,2,3,4,5,6,7,8,9,10,11,12,13,14,15,16,17] where bond breaking often occurs near the excited atomic sites. However, between the site-specific core excitation and the molecular fragmentation, there is one more step, wherein the system undergoes Auger decay. Auger decay results in a widely spread energy distribution of final states, and charge delocalization leading to different ionic fragmentation pathways. Therefore, coincidence experiments between the energy-resolved Auger electron and the fragment ions are necessary to investigate the mechanisms leading to site-specific fragmentation. We studied site-selective fragmentation of adamantane (C${}_{10}$H${}_{16}$), the smallest diamondoid molecule, using multiparticle Auger electron–ion coincidence spectroscopy and advanced theoretical calculations.

The adamantane molecule consists of a carbon cage formed by C(sp${}^{3}$)–C(sp${}^{3}$) hybridized bonds and fully terminated by hydrogen atoms, with two distinct carbon sites: methanetriyl CH and methylene CH${}_{2}$ groups. Valence dissociative ionization of adamantane has been studied using VUV radiation [18], XUV femtosecond pulses [19,20], electron ionization [21,22], and other spectroscopic techniques [23,24]. These studies have shown that the molecule tends to dissociate through multiple parallel channels that begin with the opening of the carbon cage and hydrogen migration from the CH site. In general, the fragmentation pattern of adamantane depends on the ionizing radiation. This is mainly due to the different orbitals involved in the ionizing process and the amount of internal energy deposited in the system. In our previous work [25], we found indirect evidence of site-sensitive fragmentation of adamantane after core-ionization of either the CH or CH${}_{2}$ sites. Here, we extend our research to explore site-selective fragmentation of adamantane, using resonant excitation, and compare the results to the core ionization study.

In this work, we investigated the electronic relaxation of, and the subsequent fragmentation dynamics of. adamantane at the two resonant excitations [26,27] of C 1s → valence CH orbital and C 1s → valence CH${}_{2}$ orbital at 287.1 eV and 287.6 eV, respectively. We modeled theoretical X-ray absorption spectra via single point and nuclear ensemble approaches to understand the nature of these excitations. We used theoretical calculations within the EOM–CCSD framework to interpret the measured resonant Auger electron spectra (RAES). Additionally, we predicted the nuclear dynamics in the core excited state of adamantane using the ‘Z + 1 approximation’. We find clear site-dependent bond dissociation for the dominant fragmentation channels of the adamantane cation, proving site sensitivity to the initial core–hole localization, either on the CH or CH${}_{2}$ carbon site. Interestingly, we noticed a significant decrease in the production of methylium (CH${}_{3}^{+}$) when core-excited adamantane was compared to core-ionized adamantane. This finding suggests the existence of new mechanisms for cage-opening processes that do not involve hydrogen migration or CH${}_{3}$ termination.

## 2. Results and Discussion

### 2.1. Resonant Excitation of Adamantane

Figure 1a presents the experimental total ion yield (TIY) and the theoretical XAS spectrum of adamantane below the C 1s ionization edge. We observed two sharp peaks at 287.1 eV and 287.6 eV, previously reported by Willey et al. [26,27], and assigned to C 1s → valence CH orbital and C 1s → valence CH${}_{2}$ orbital transitions respectively. The theoretical XAS spectra were calculated at the CVS-EOM-EE-CCSD/6-31g* level, via the single point approach, and CAM-B3LYP/6-31g*, via the NEA approach. Both the theoretical XAS spectra were congruent with the experimental TIY spectral shape and spacing between the first and second peaks (other levels of theory and discussions are available in the Appendix A). The experimental spectrum exhibited vibronic structures that were neglected in the calculations. According to the calculations, the lower energy peak at 287.1 eV originates from resonant transitions involving both CH and CH${}_{2}$ 1s carbon sites to their respective valence orbitals, with the contribution from the CH site being predominant (see Appendix A). The higher energy peak at 287.6 eV corresponds, specifically, to transitions from CH${}_{2}$ 1s carbon sites to CH${}_{2}$ valence orbitals. By tuning the photon energy to 287.1 eV or 287.6 eV, we can primarily excite the CH or the CH${}_{2}$ carbon sites in adamantane, respectively.

After the core resonant excitation, the interplay between nuclear dynamics and electronic relaxation becomes crucial. We investigated the nuclear dynamics of adamantane following core resonant excitation using the equivalent core model [16,28]. The evolution of the molecular geometries in the core-excited states for the CH and CH${}_{2}$ carbon sites are shown in Figure 1b,c, respectively. In the Z + 1 approximation, when a localized core–hole is created, the valence electrons in the molecule experience an effective nuclear charge that increases by one, which induces changes in bond distances, angles, and potential surfaces. We observe in Figure 1b that a localized CH core–hole state induced a change in the symmetry of adamantane from T${}_{d}$ to C${}_{3v}$, which eventually led to the loss of one hydrogen atom. One the other hand, a localized CH${}_{2}$ core–hole state induced a change in symmetry of adamantane from T${}_{d}$ to C${}_{2v}$, which elongated one C–H bond, which further changed to C${}_{s}$ symmetry and, eventually, led to loss of one hydrogen atome. Therefore, the localized core holes at the CH and CH${}_{2}$ sites of adamantane exhibited distinct evolution patterns and significant distortion in the molecular geometry, which influence the subsequent Auger decay process and the fragmentation dynamics.

### 2.2. Resonant Auger Decay of Adamantane

Figure 2 shows the resonant Auger electron spectra (RAES) of adamantane for inner-shell resonant excitation to the CH (orange) and CH${}_{2}$ (purple) valence orbitals lying at photon energies of 287.1 and 287.6 eV, respectively. The RAES is presented in the binding energy scale, calculated as the difference between the photon energy and the auger electron kinetic energy. In the RAES, the final states of the adamantane cation [29,30] (grey dashed lines) are assigned using the reference photo-emission spectrum of valence ionization (black) taken at an off-resonance photon energy of 285 eV. Two distinct regions were identified in the RAES, corresponding to participator Auger decay (1h) at the low binding energy (<17.5 eV), and spectatorAuger decay (2h-1p) at the high binding energy (>25 eV). By comparing to the off-resonant and the normal Auger spectra reported in our previous study [25], the region in the vicinity of the IT${}_{2}$ is referred to as a *mixed* region, due to possible overlap of 1h and 2h–1p states [31,32,33]. The final Auger-populated states lying above the double ionization threshold (IT${}_{2}$) at 23.9 eV auto-ionize to form adamantane dication. We observed that the CH site excitation was more likely to decay via participator, or mixed, Auger decay, whereas the CH${}_{2}$ site excitation was more likely to decay via spectator Auger decay. The strongest contribution in the RAES of the CH sites was observed at around 24.4 eV, and of the CH${}_{2}$ sites at around 31.2 eV.

For binding energies below 25 eV, we calculated the theoretical participator Auger decay using the same level of theory as the XAS. The calculated participator Auger decay spectra at 287.1 eV and 287.6 eV are shown in Figure 3, with the experimental RAES. In the participator decay region below <17.5 eV, the theoretical RAES was in good agreement with the peak intensities of the 1h states in the experimental RAES at the two carbon sites. In the mixed decay region, however, the theoretical RAES only estimated the intensity of 1h states and was not able to replicate the mixed character of decay. We observed, in the experimental and theoretical RAES, that the probability of reaching states with t${}_{2}$ symmetry was generally higher after CH excitation, suggesting a possible interplay between this electronic symmetry and the localization of the hole on the C–H bond. This hypothesis was corroborated by the difference between the spectra populating the cationic ground state, 7t${}_{2}^{-1}$, where the hole is reported to be localized on the C–H bond [34]. The theoretical RAES was calculated with frozen geometries and did not calculate the nuclear dynamics in the core-excited state. However, in the experimental RAES, the maximum of many peaks shifted towards higher binding energy compared to the off-resonance spectrum. This reflected the nuclear wavepacket dynamics occurring in the core-excited states, such as those predicted by the Z + 1 approximation, which, after electronic relaxation. allow the reaching of potential energy surfaces outside the Franck–Condon region. Such nuclear dynamics lead, among other things, to vibrational excitation [35,36,37] and seem more pronounced after CH excitation than after CH${}_{2}$ excitation. Therefore, we found a significant difference between the RAES of the CH and CH${}_{2}$ sites. Our experimental and theoretical results showed that the “memory” of the localization of the initial core hole in adamantane was partly preserved after resonant Auger decay.

### 2.3. Site-Selective Fragmentation of Adamantane

We studied the fragmentation of adamantane ions after resonant Auger decay using Auger-electron photo-ion coincidence (AEPICO) spectroscopy. Our results, presented in Figure 4a,b, show that the fragment ions for both the CH and CH${}_{2}$ sites were similar, consisting of a series of hydrocarbon fragments, C${}_{n}$H${}_{x}^{+}$, where x was mainly an odd number. The dominant ions for both sites were C${}_{3}$H${}_{3}^{+}$, C${}_{2}$H${}_{3}^{+}$, and C${}_{3}$H${}_{5}^{+}$, with the number of hydrogen atoms attached depending on the binding energy of the final Auger-populated state. This trend was clearly visible for the C${}_{4}$H${}_{3}^{+}$, C${}_{4}$H${}_{5}^{+}$ and C${}_{4}$H${}_{7}^{+}$ fragments in the AEPICO maps (Figure 4a,b). Our findings suggest that the final state energy influences hydrogen migration or evaporation during fragmentation. Additionally, for both sites, symmetric fragmentation of the carbon cage to form C${}_{5}$H${}_{x}^{+}$ fragments was unfavourable for all the final states. Figure 4c highlights the site-selectivity in fragmentation, showing the difference between the AEPICO plots for the CH and CH${}_{2}$ sites. The areas in red had a higher yield for the CH site, while the areas in blue had a higher yield for the CH${}_{2}$ site. Our data indicated that, after CH site excitation, the adamantane ions were more likely to dissociate into larger fragments, such as C${}_{6}$H${}_{7}^{+}$, C${}_{6}$H${}_{9}^{+}$, and C${}_{7}$H${}_{9}^{+}$, while the CH${}_{2}$ site excitation resulted in dissociation into smaller fragments, such as C${}_{3}$H${}_{3}^{+}$ and C${}_{3}$H${}_{5}^{+}$. These differences in fragmentation appeared at different binding energies, suggesting that the site-selective Auger decay created statistical bias in the fragmentation pattern, with CH site excitation preferentially populating the adamantane cationic states and CH${}_{2}$ site excitation resulting in Auger-populated dicationic states. In such a case, the fragmentation was actually state-specific, as previously reported for 2Br-pyrimidine [14]. The fragments C${}_{2}$H${}_{3}^{+}$, C${}_{3}$H${}_{3}^{+}$, C${}_{3}$H${}_{5}^{+}$, C${}_{6}$H${}_{7}^{+}$, and C${}_{7}$H${}_{9}^{+}$ had significant intensities in the difference AEPICO map (Figure 4c) and were, therefore, selected for further analysis of site-selectivity.

In order to gain a deeper understanding of the fragmentation behavior of adamantane, we investigated whether final Auger-populated states with identical binding energies, resulting from excitation of the CH and CH${}_{2}$ sites, exhibited discernible characteristics during fragmentation. The Auger electron energy-selected yield of ions in the AEPICO plot for the CH and CH${}_{2}$ site excitations are presented in Figure 5a,b, respectively. These plots depict the relative yield of the selected ions as a function of the binding energy of the final Auger-populated states. The Auger-populated states located far below IT${}_{2}$ represent the yield of cation fragmentation, while those close to, and above, IT${}_{2}$ indicate the yield of dication fragmentation.

The fragmentation pattern of adamantane is influenced by the binding energy of the Auger-populated states, and the dominant fragment ion exhibits site-selectivity for states with binding energies lower than 30 eV. The dominant ion at lower binding energy states was C${}_{6}$H${}_{7}^{+}$, but it transitioned to C${}_{3}$H${}_{5}^{+}$ for states above 20 eV, and then to C${}_{3}$H${}_{3}^{+}$ for states above 30 eV, for both the CH and CH${}_{2}$ sites. This transition in the yield of C${}_{6}$H${}_{7}^{+}$ and C${}_{3}$H${}_{5}^{+}$ was indicative of the onset of dication fragmentation [19] and suggested that the states above 20 eV underwent auto-ionization. The relative yields of C${}_{6}$H${}_{7}^{+}$ and C${}_{3}$H${}_{5}^{+}$ were site-selective and state-specific. In the case of CH site excitation, we observed that cationic states preferentially fragmented into C${}_{6}$H${}_{7}^{+}$, with a neutral fragment C${}_{4}$H${}_{9}$. The yield of this fragmentation channel was highest (20.5%) at a binding energy of about 16 eV and could be correlated to the nearly degenerate 5t${}_{2}$,5a${}_{1}$${}^{-1}$ states. In contrast, in the case of CH${}_{2}$ site excitation, the relative yield showed that cation fragmentation could lead to either C${}_{6}$H${}_{7}^{+}$ or C${}_{3}$H${}_{5}^{+}$ fragments. The yield of C${}_{6}$H${}_{7}^{+}$ was only slightly enhanced, with a maximum (11%) at 18.5 eV, which could be correlated to the 4t${}_{2}$${}^{-1}$ state. Additionally, the yield of C${}_{7}$H${}_{9}^{+}$ was also site-selective. For the CH site, the yield was highest (∼8%) in the range of 12 to 18 eV, but for the CH${}_{2}$ site, the yield was highest at 12.6 eV and decreased drastically at higher binding energies. These results suggested that cationic states reached after participator Auger decay displayed site-selectivity, and that the memory of the core–hole location influenced cationic fragmentation to some extent. In contrast, for spectator Auger-populated states, the initial core–hole location did not affect the fragmentation pathway, and the dications produced mainly C${}_{3}$H${}_{3}^{+}$ and C${}_{2}$H${}_{3}^{+}$ ions for both sites. The difference in the yield of C${}_{3}$H${}_{3}^{+}$ was a result of site-dependent Auger decay, which led to a higher probability of reaching favourable final states after CH${}_{2}$ site excitation. However, when the relative yield of ions at these higher binding energy states was examined, the yield of C${}_{3}$H${}_{3}^{+}$ was the same for both sites.

The difference in the yield of C${}_{6}$H${}_{7}^{+}$ and C${}_{7}$H${}_{9}^{+}$ was attributed to the distinct properties of the final Auger-populated states with the same binding energy, which dissociated differently for the two sites, with a preference for the CH site. The maximum yield of these fragments could be correlated to the nearly degenerated 5t${}_{2}$/5a${}_{1}$ states, while their yields vanished after the 4t${}_{2}$ states. In comparison, these fragmentation channels were less favourable after CH${}_{2}$ excitation. The presence of states with t${}_{2}$ symmetry was in line with our electron spectroscopy observations and supported the idea that some states with this electronic symmetry reflect the localization of a hole on a C–H bond. The localization of a hole on the C–H bond leads to distortion of the adamantane cation [34], involving elongation of three C–C bonds and the formation of C${}_{6}$H${}_{9}$ and C${}_{4}$H${}_{7}$ units, in concurrence with the observation of the C${}_{6}$H${}_{7}^{+}$ fragment. However, other factors, such as an increase in the internal rovibrational energy, must be taken into consideration to account for the disappearance of these fragments. Such factors can cause the emission of neutral hydrogen atoms or the cleavage of C–C bonds. It should be noted that these fragments have been shown to reveal information on the cage-opening mechanisms at lower excitation energies [18], suggesting that the cage-opening mechanism of the adamantane cation is site-sensitive.

### 2.4. Comparision to Core-Ionized Adamantane

The mass spectrum of ions produced from the fragmentation of adamantane dication following core excitation of the CH and CH${}_{2}$ sites and core ionization is shown in Figure 6. The mass spectrum was calculated as projections of Auger-electron Photoion–Photoion coincidence (AE-PIPICO) maps. In our previous AE-PIPICO study [25], we speculated that site-sensitive fragmentation of adamantane dication after core-ionization indirectly used theoretical normal Auger decay calculations. Here, we directly studied site-selective fragmentation using core excitation. We observed that, for both the CH and CH${}_{2}$ sites, the mass spectrum was rather similar, with dominant ions C${}_{3}$H${}_{3}^{+}$, C${}_{2}$H${}_{3}^{+}$ and C${}_{3}$H${}_{5}^{+}$. Therefore, the fragmentation of the adamantane dication did not exhibit site selectivity and likely depended only on the internal energy of the dicationic state. However, the fragmentation of dication is different following core excitation and core ionization. In the core ionization case, the dominant ions are C${}_{2}$H${}_{3}^{+}$, C${}_{3}$H${}_{3}^{+}$, C${}_{3}$H${}_{5}^{+}$ and CH${}_{3}^{+}$ with broad peaks, due to higher kinetic energy release compared to the core excitation case.

Maclot et al. [20] showed that the fragmentation of adamantane dication involves barrier-free structural changes, such as hydrogen migration(s) or cage-opening, before dissociating into different ion pairs C${}_{n}$H${}_{x}^{+}$/C${}_{m}$H${}_{y}^{+}$. The hydrogen migrations result in the open-cage geometry of adamantane, with CH${}_{3}$ termination(s). The lowest energy open-cage structure of adamantane dication is shown in blue. In comparison to core-ionization, the cage opening mechanism seems fundamentally different for core excitation of the CH and CH${}_{2}$ sites, since the intensity of the methylium (CH${}_{3}^{+}$) ion (peak highlighted with a red arrow) decreases by 44% in the AE-PIPICO mass spectrum. Within the Z + 1 approximation, we found that the symmetry broke after CH and CH${}_{2}$ excitation and could lead to hydrogen loss in the core–hole state. During the lifetime of the core–hole, ultrafast nuclear dynamics, mainly of hydrogen atoms, contribute to a distortion of the adamantane cage. In such cases, the calculations reported by Maclot et al. [20] would not be valid for our core excitation studies. One could speculate that rather large transient species are produced; for instance, containing 4 and 6 carbons, instead of the intact open-cage geometry. Nevertheless, there is a difference in the fragmentation of the adamantane dication after core excitation compared to core ionization. This difference may be attributed to the distinct ultrafast dynamics that occur in the core–hole and dication states leading to the fragmentation.

## 3. Methods

### 3.1. Experimental

The experiments were conducted at PLEIADES soft X-ray beamline at Synchrotron SOLEIL [38]. The Apple II HU 80 permanent magnet undulator generated the soft X-rays, which were monochromatized using the modified Petersen plane grating monochromator, with a high-flux 600 lines per mm grating. The CO${}_{2}$ C-1s → π* absorption peak [39] was used to calibrate the photon energy for the present experiment. A heated gas cell [25] was used, with a VG-Scienta R4000 electron energy analyzer, to perform electron spectroscopy measurements [40]. The high-purity commercial adamantane powder (99%, Sigma-Aldrich, Saint-Quentin-Fallavier, France) was used as a sample without further purification. During the measurements, the gas cell was heated at 54 degrees Celsius, and the pressure in the Scienta chamber was 7.1 × 10${}^{-7}$ mbar. The X-ray absorption spectrum was taken with the total ion yield technique and a resolution of 25 meV. The kinetic energy scale of the adamantane Auger spectra was calibrated according to the measured CO${}_{2}$ Auger spectrum and the position of reference lines reported by Modeman et al. [41]. The Auger electron spectra of adamantane were recorded at a photon energy of 287.1 and 287.6 eV with an overall resolution of 55 meV.

The EPICEA setup [15,42], consisting of a double toroidal electron analyzer (DTA) [43] and a 3D focusing ion TOF spectrometer, was used to record the Auger-electron photo-ion coincidence (AEPICO) and Auger-electron photoion–photoion coincidence (AE-PIPICO) data. Electrons emitted at an angle of 54.7° were retarded to a predetermined pass energy (Ep) in the DTA. The detection electron kinetic energy window was about 12% of the defined Ep, whereas the energy resolution was about 1% of the Ep. In the present experiments, a pass energy of 250 eV was used, which led to an analyzer resolution of about 2.5 eV and a detection range of ±15 eV. The calibration of the kinetic energy scale was performed using Xe 5p and Xe 5s IPs as reference lines. The kinetic energy scale calibration was achieved by using a previously proposed empirical formula [44] to transform the detector radius position (mm) to the electron kinetic energy (eV). The adamantane was introduced into the vacuum chamber through a heated injection gas line and a heated needle in a crossed-beam experimental arrangement. In the EPICEA setup, the detection of an electron triggers a pulsed field that accelerates all ions toward a 3D momentum mass spectrometer. Hence, a coincidence event consists of an energy analyzed electron, positions on the detector and time of flight of all extracted ions. Additionally, a pulse generator was used to produce random triggers, which extracted ions present in the interaction. This allowed us to subtract the background ion signal, the *false* coincidences, using the procedure reported by Prumper et al. [45].

### 3.2. Theoretical

The X-ray absorption spectra (XAS) were modeled at the CVS-EOM-CCSD level with 6-31g* and 6-31+g* basis sets for the structure optimized in the ground state at the CAM-B3LYP/6-31+g* level. Furthermore, the spectra were modeled at various TDDFT levels, via the nuclear ensemble approach (NEA). The character of the excitations was studied via natural transition orbitals. The RAES were modeled in terms of the Feschbach–Fano theory [46,47] at the EOM-CCSD level with 6-31g* and 6-31+g* basis sets, as implemented in Q-Chem [48]. All details are provided in the Appendix A.

We also performed simulations in the frame of the Z + 1 approximation. To do this, we considered the initial geometry that optimized the neutral adamantane in the ground state was at the B3LYP/6-31g* level. Then, substituting one C atom with one N atom, and keeping a neutral charge, we explored the potential energy surface, allowing the geometry to relax, following the minimum energy path (downhill) up to a minimum being reached, using the same level of theory—B3LYP/6-31g*. These calculations were performed with the Gaussian16 program [49].

## 4. Conclusions

We studied the fragmentation dynamics of adamantane after resonant inner-shell excitation using a state-of-the-art coincidence technique between Auger electrons and ions in combination with advanced theoretical calculations. The resonant Auger decay spectra provided evidence, and, in particular, the electronic states with a t${}_{2}$ symmetry, of there being memory of the localization of the initial core–hole upon resonant excitation to the CH or CH${}_{2}$ valence orbitals. After *participator* Auger decay, the fragmentation of the molecules leading to large fragments with 6–7 carbon atoms exhibited a clear site-sensitivity of the initial core–hole localization, proving that different carbon sites could lead to different cage opening mechanisms. After *spectator* Auger decay in the double ionization continua, the fragmentation dynamics was drastically different to that observed after core–hole ionization. The absence of the methyl group, in coincidence with a large fragment, suggested that the cage opening mechanisms led to rather large short-lived species, which eventually underwent further fragmentation. 

## Figures and Tables

**Figure 1 molecules-28-05510-f001:**
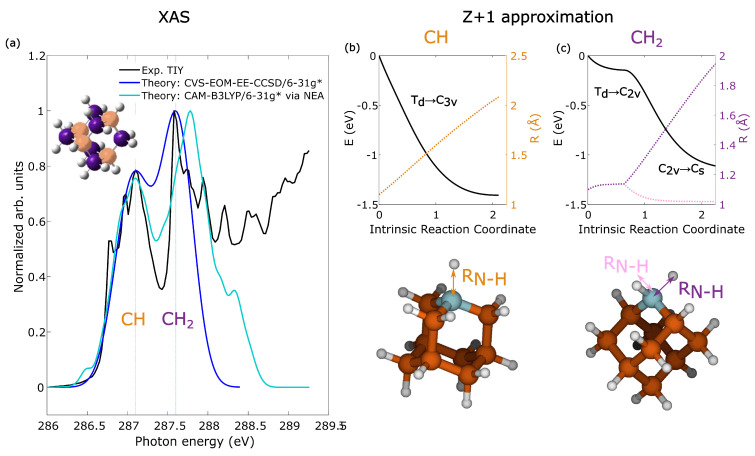
The experimental total ion yield (TIY) and the theoretical X-ray absorption (XAS) spectrum of adamantane below the C 1s ionization edge is shown in (**a**). The theoretical spectra were shifted to match the experiment, −3.60 eV for CVS-EOM-CCSD and 8.97 eV for CAM-B3LYP. The ground state carbon cage structure of the adamantane molecule with the highlighted CH (orange) and CH${}_{2}$ (purple) sites is shown in the top left. The results from the equivalent core model [28] calculation with Z + 1 approximation for localized core holes at the CH and CH${}_{2}$ carbon sites are shown in (**b**) and (**c**), respectively. The final calculated distorted geometry of the adamantane is shown below for the two carbon sites.

**Figure 2 molecules-28-05510-f002:**
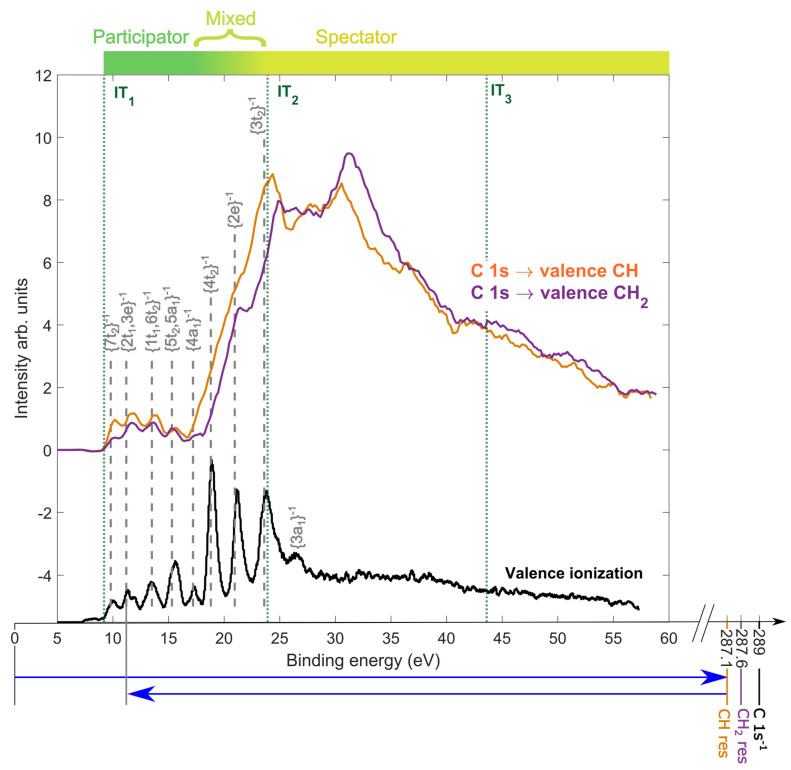
Resonant Auger electron spectra (top) of adamantane after CH and CH${}_{2}$ resonant excitation at 287.1 eV and 287.6 eV, respectively, are shown in orange and purple, respectively. The off-resonance photoelectron spectrum (bottom) of valence ionization of adamantane at 285 eV is also shown in black. The assigned cation states [29,30] after participator Auger decay are shown in grey. Three distinct regions were identified in the spectrum, corresponding to participator, mixed and spectator Auger decay. The first, second and third ionization thresholds were labelled as (IT${}_{1}$), (IT${}_{2}$) and (IT${}_{3}$), respectively. At the bottom, a simple energy level diagram shows the resonant excitations and Auger decays to final states (for instance the {2t${}_{1}$/3e}${}^{-1}$ cation state).

**Figure 3 molecules-28-05510-f003:**
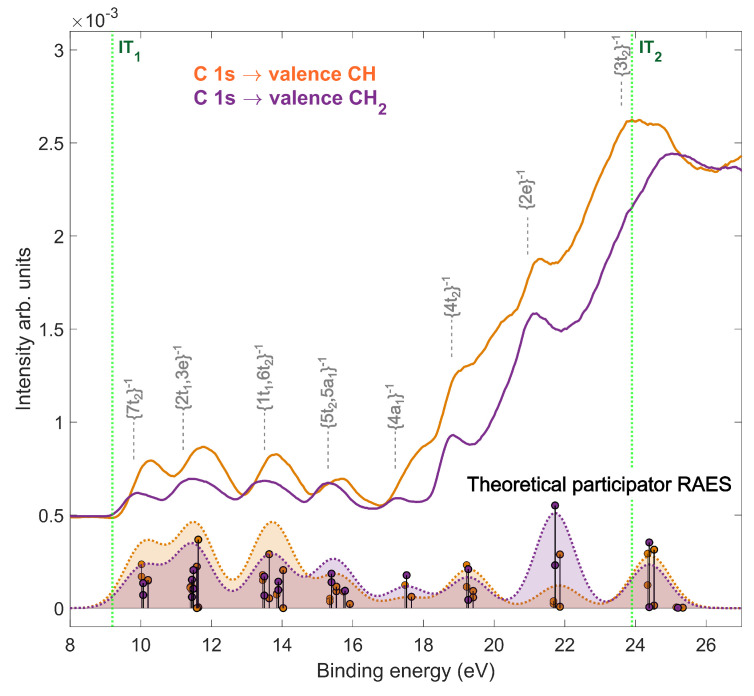
Experimental (top) and theoretical (bottom) resonant Auger electron spectra (RAES) of adamantane after CH and CH${}_{2}$ resonant excitation at 287.1 eV and 287.6 eV, respectively, are shown in orange and purple, respectively. The theoretical spectra were shifted by 4.1 eV to match the experiment. The assigned cation states [29,30] after participator Auger decay are shown in grey. The first and second ionization thresholds were labeled as (IT${}_{1}$) and (IT${}_{2}$), respectively.

**Figure 4 molecules-28-05510-f004:**
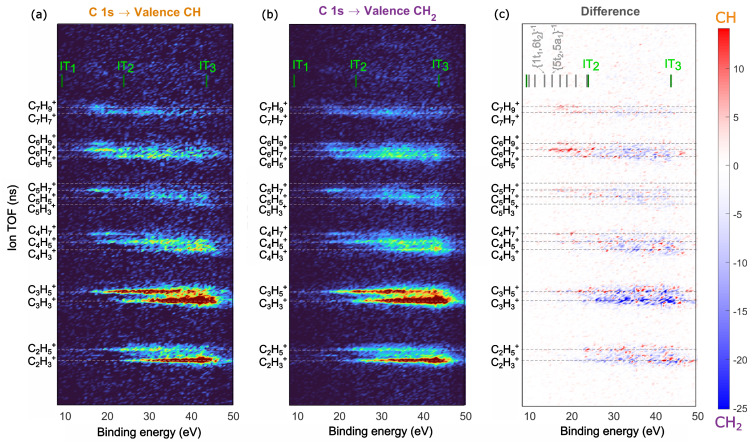
AEPICO map of ions and auger electrons detected in coincidence for the CH (**a**) and CH${}_{2}$ (**b**) resonant excitations at 287.1 eV and 287.6 eV, respectively. The difference between the AEPICO maps (**a**,**b**) is shown in (**c**), the intensity in red indicates ions which had higher yield in (**a**) i.e., the CH site excitation and the intensity in blue indicates ions which had higher yield in (**b**) i.e., the CH${}_{2}$ site excitation. The first, second and third ionization thresholds are labeled as (IT${}_{1}$), (IT${}_{2}$) and (IT${}_{3}$), respectively. The maps (**a**,**b**) are area normalized.

**Figure 5 molecules-28-05510-f005:**
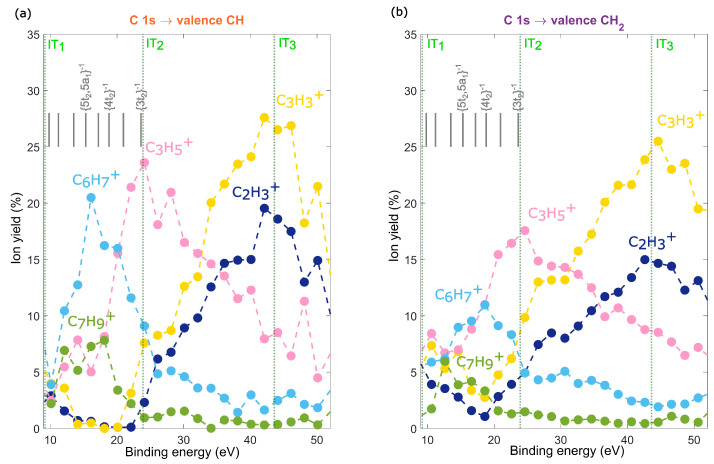
The energy-selected relative branching ratios of ions for the CH (**a**) and CH${}_{2}$ (**b**) resonant excitations at 287.1 eV and 287.6 eV, respectively. The relative branching ratios were calculated from the AEPICO map by selecting the binding energy of the Auger-populated states. The assigned cation states after participator Auger decay are shown in grey. The first, second and third ionization thresholds are labeled as (IT${}_{1}$), (IT${}_{2}$) and (IT${}_{3}$), respectively.

**Figure 6 molecules-28-05510-f006:**
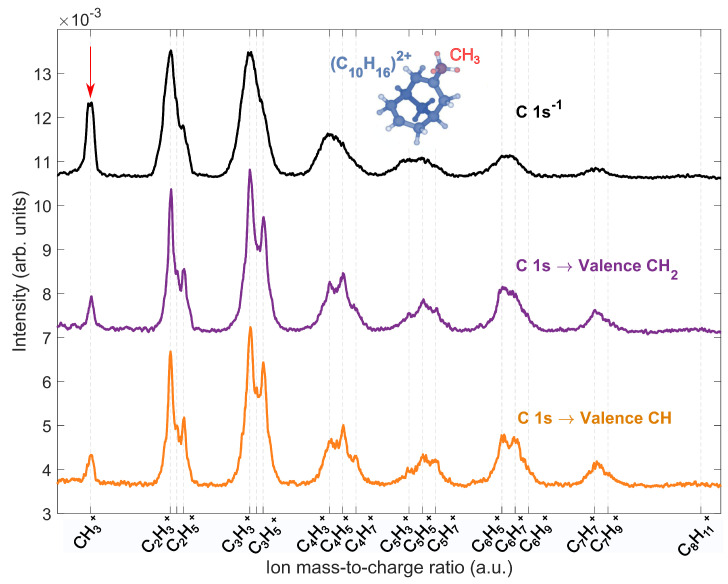
The mass spectrum of adamantane dication fragmentation after CH (orange) and CH${}_{2}$ (purple) resonant excitations at 287.1 eV and 287.6 eV, respectively, and core ionization (black) at 350 eV. The mass spectrum was added up from projections of the 2D AE-PIPICO map. The lowest energy open-cage geometry [20] of adamantane dication with one CH${}_{3}$ termination is shown in blue. The difference in the intensity of the methylium (CH${}_{3}^{+}$) peak is highlighted with a red arrow.

## Data Availability

The data presented in this study are available on request from the corresponding author.

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
