# Peer review of "Resonant Inner-Shell Photofragmentation of Adamantane (C10H16)"

_molecules, 2023, doi:10.3390/molecules28145510_

Round 1
Reviewer 1 Report
This manuscript studied the resonance inner-shell photofragmentation of the adamantane with experimental and computational method. The experiment was carried out well, the conclusion is sound, and the manuscript writing is well organized. I recommend acceptance of the manuscript for publication in its current form.
Author Response
A point-by-point response to the reviewers' comments is given in the attached PDF file.

Reviewer 2 Report
Review Report:
In their manuscript, Ganguly et al., demonstrated electronic relaxation and the following photofragmentation dynamics of adamantane after inner shell resonant excitation at 287.1 eV and 287.6 eV, which correspond to C → valence CH and C → valence CH2 orbitals, respectively. These transitions were interpreted via theoretical XAS calculations and the measured resonant Auger electron spectra were also modelled. They successfully identified site dependent preference for different dissociation channels. In addition, the major dissociation product, CH3+, resulting from core-ionization is substantially reduced in the case of resonant excitation.
I find the work important in the context of understanding different fragmentation channels resulting from various modes of inner shell excitation and/or ionization. The manuscript is thorough and well-written. I recommend publication of the present work essentially as is. I have minor comments. I used the following abbreviations, P-page number and L-line number to point out my comments.
Comments:
1. P3-Figure 1: In the caption, there is a repetition of the word “state”.
2. P3-L100: I am curious to know about the source of shoulder peaks below the 287.1 eV transition in the experimental TIY spectrum.
3. I encourage the authors to provide a MO diagram mentioning the inner shell (i.e., C 1s) and the valence orbitals and the corresponding transitions for better readability.
4. In the supporting material under the section valence band XPS section, the 1st sentence seems to me incomplete.
Author Response

(The authors gave the same response as above.)

Reviewer 3 Report
The authors reported on the fragmentation dynamics of C10H16 after resonant inner shell excitation experimentlly and theoretically. This piece of work is an extension of their recent publications. Substantial rework is strongly recommended before considering publication.
1. The title: “Adamantane” could be replaced by C10H16, I invite then the authors to slightly change the title.
2. Fig. S1. The FWHMs were 0.2 eV and 0.05 eV, respectively. Why are they different? What is the physical reason? “Spectra were shifted to match the position”, what are exact values? Same for different calculation methods?
3. Authors mentioned the molecular dynamics simulation, however, no simulation data or results can be visited.
4. Fragmentation dynamics of C10H16 is a well-studied system. The loss channels are quite different under different experiment. Authors are suggested to analyze and comment on the main reason on dissociate channels.
5. I would appreciate the future publication of a more complete paper, instead of many fragmented papers by the authors.
Author Response

(The authors gave the same response as above.)

Round 2
Reviewer 3 Report
The manuscript by Ganguly has been revised, but it is not enough, at least to me, for the manuscript to be published. Further revision based on the following points is needed.1. about the title, perhaps "Adamantane (C10H16)" more suitable, but it is only a minor problem. Authors could make their owen dicision.
2. Authors added the shifted values for different calculations. We see the values are quite different and calculted spectra are quite different also (all are DFT methods, can be understand). Two more questions: a) is that necessary to present the results for 6-31g*? b)is that difficult to perform additional CCSDT/CISDT/.. calculations?
Author Response
The reply is given in the attached PDF file.
